Assay of cardiopulmonary bypass system for porcine alveolar macrophages removing GFP-E. coli from erythrocyte surfaces

Liu Yongqiang
Wang Nan
Ru Qing
Fan Kuohai
Sun Na
Sun Panpan
Li Hongquan
Yin Wei dkyyinwei@126.com
Shanxi Key Laboratory for Modernization of TCVM, College of Veterinary Medicine, Shanxi Agricultural University , Jinzhong City, Shanxi Province , China
Haraguchi Tokuko
Electronic publication date: 2025 Mar 4
Publication date: 2025
Volume: 13
Electronic Location ID: e18934
Received 2024 Sep 26; Accepted 2025 Jan 14
Copyright: © 2025 Liu et al.
Copyright year: 2025
Copyright holder: Liu et al.
License: This is an open access article distributed under the terms of the Creative Commons Attribution License, which permits unrestricted use, distribution, reproduction and adaptation in any medium and for any purpose provided that it is properly attributed. For attribution, the original author(s), title, publication source (PeerJ) and either DOI or URL of the article must be cited.
License URL: https://creativecommons.org/licenses/by/4.0/

Keywords: CR1-like, FcR, GFP-E. coli, PAMs, Porcine erythrocytes

Funding: Shanxi Agricultural University Doctoral Scientific Research Initiation Project 2021BQ77 Shanxi Natural Science Research Programme Project 20210302123407 Shanxi Key lab for Modernization of TCVM 202104010910015 This work was supported by the Shanxi Agricultural University Doctoral Scientific Research Initiation Project (2021BQ77), the Shanxi Natural Science Research Programme Project (20210302123407) and the Shanxi Key lab for Modernization of TCVM (202104010910015). The funders had no role in study design, data collection and analysis, decision to publish, or preparation of the manuscript.

==============================
While it is established that complement receptor molecules on the surface of erythrocytes are crucial for the clearance of immune complexes in the body, the molecular mechanisms underlying the interaction between macrophages and erythrocytes in pigs remain inadequately understood. Consequently, we built a detection system with a closed-circulation flow chamber and a constant flow pump. Additionally, we optimized parameters including system flow velocity and fluid shear force. In the circulatory system, our study measured the fluorescence intensity of erythrocyte and pulmonary alveolar macrophages (PAMs) surfaces before and after the blockade of complement receptor 1 (CR1)-like receptors and Fc receptors. The results indicated that porcine erythrocytes and PAMs exhibited a diminished rate of change in fluorescence intensity under the blocked condition. Through transmission electron microscopy, it was observed that PAMs effectively removed sensitized GFP-E. coli adhering immunologically to porcine erythrocytes. The findings indicate that PAMs effectively removed sensitized GFP-E. coli from the surface immunoadhesion of porcine erythrocytes, facilitated by the mediation of surface CR1-like receptors and Fc receptors.

Introduction

Research in the medical field has demonstrated that the excessive accumulation and abnormal removal of immune complexes (ICs) within the body can contribute to the onset and development of various immune-related diseases, including glomerulonephritis (Kojima & Oda, 2022), systemic lupus erythematosus (Kiriakidou & Ching, 2020), Alzheimer’s disease (Scheltens et al., 2021) and rheumatoid arthritis (Kremlitzka et al., 2013). Numerous studies have demonstrated the presence of complement receptor 1 (CR1) on the surface of erythrocyte membranes, facilitating the transport of erythrocyte adhesion immune complexes (ICs) to the mononuclear phagocyte system for clearance (Al-Saleem et al., 2017; Birmingham & Hebert, 2001). In non-primate mammals, red blood cells also perform immune adhesion functions, although the CR1 on red blood cells in primate mammals has been the subject of more extensive research (Chen et al., 2000; Li et al., 2010; Zhu et al., 2011). Research on the immunoadhesion receptors of erythrocytes in various animal species has advanced more gradually. During the initial laboratory phase, it was verified that porcine erythrocytes possess immunoadhesive function, and have characterized a CR1-like immunoadhesion receptor on porcine erythrocytes (Sun et al., 2012; Yin et al., 2015). The co-culture experiments involving immune complex (IC)-bound human erythrocytes and mouse macrophages have demonstrated that ICs were detectable within the mouse macrophages (Reinagel & Taylor, 2000). This finding suggests that erythrocyte-bound ICs are capable of being phagocytosed and internalized by the mouse macrophages. Therefore, an in-depth examination of the molecular mechanisms underlying macrophage-erythrocyte interactions are essential to elucidate the biological roles of porcine erythrocytes and their surface complement receptor molecules in the context of animal diseases (Yin et al., 2019; Chun, 2019).

Materials

Animals and bacterial strains

Three healthy Landrace pigs, weighing 20 ± 2 kg, were purchased from Xinsihai pig farm in Wuxiang County, Shanxi Province, China. Healthy rabbits, weighing 2 ± 0.5 kg, were purchased from Shanxi Agricultural University Animal Production Laboratory. The acquired test animals were housed in a cleaned, well-ventilated isolated room for a 7-days acclimatization period without any experimental intervention. During this period, they were provided with nutritionally balanced feed and clean drinking water. A strain of E. coli expressing green fluorescent protein (GFP-E. coli) was utilized in this study. All animals used in the present experiments were cared for humanely and the use of the animals was approved by the Animal Ethics Committee at the Veterinary Medicine College of Shanxi Agriculture University in China (No. SXAU-EAW-2022P.ST.008003209).

Main reagents and equipments

The main reagents include mouse anti-porcine CR1-like monoclonal antibody (Patent No. ZL201410308534.0, prepared by our laboratory); mouse anti-human FcR McAb (Shanghai Shanjin Biotechnology Co., Ltd., Shanghai, China); RPMI-1640 medium (Gibco, Waltham, MA, USA); fetal bovine serum (Bioland, Los Angeles, CA, USA); porcine lymphocyte isolate, (Tianjin Hao Yang Biotechnology Co., Ltd, Tianjin, China). The main equipments used included CO2 incubator (Shanghai Lishen Scientific Equipment Co. Ltd., Shanghai, China); −80 °C ultra-low temperature refrigerator (Zhongke Meiling Cryogenic Technology Co., Ltd., China); peristaltic pump (Lange Constant Flow Pump Co., Ltd., Baoding, China); Laminar Flow hood (SW-CJ-2FD, Airtech, Ltd., Suzhou, China); BD FACSCalibur flow cytometer (BD Bioscience, Franklin Lakes, NJ, USA); inverted fluorescence microscope and Biological Transmission Electron Microscopy (Olympus Corporation, Shinjuku, Tokyo, Japan). The Parallel-Plate Flow Chamber CR@3, which measures 7.5 cm in length, 1 cm in width, and 0.5 cm in height, is accompanied by packaging that includes a sealing washer and a connecting silicone hose. These components were procured from the GlycoTech company.

Methods

The preparation of cells

Two milliliters of sterile blood were collected from the anterior vena cava of piglets that had been fasted for 8 h, using a sterile 5 ml syringe. Following the weighing of the piglets, a specially designed piglet squeeze frame was applied to position them on their left side. Anesthesia was then administered via the ear margin vein, using a combination of ketamine at a dosage of 20 mg/kg and xylazine at 2 mg/kg. Once deep anesthesia was achieved, a scalpel was used to incise the carotid artery, and the right carotid artery was bluntly dissected. A catheter was subsequently inserted into the artery to facilitate bloodletting. The criteria for determining the death include cardiac arrest, respiratory arrest, and mydriasis. In the cell chamber, sterile porcine whole lungs were lavaged with phosphate-buffered saline (PBS) at pH 7.4, filtered, centrifuged, washed, and suspended with PBS. The cell density was adjusted to 1.2 × 107/mL and designated as sample I. Utilizing the porcine red blood cell separation kit, 2 mL of whole blood was mixed with an equal amount of diluent, added the separation solution, centrifuged, and resuspended with PBS. The cell density was adjusted to 1.2 × 107/mL and designated as sample II. Ten milliliters of blood from a healthy rabbit were collected in the animal isolation room for the purpose of serum preparation. Porcine erythrocytes were then co-incubated with serum-sensitized GFP-E. coli, and this preparation was designated as Sample III.

The flow circulation chamber device

The flow chamber washer was positioned at the periphery of the slide, and connections were established between the flow chamber, the mobile phase storage bottle, and the peristaltic pump using a silicone hose. The 20 mL aliquot of Cell Suspension Sample III was transferred into a mobile phase storage vial. The peristaltic pump was then activated to facilitate the circulation of the suspension within the mobile phase reservoir, ensuring complete filling of the entire system. The speed of the peristaltic pump was adjusted according to the specifications outlined, and the volume of the collected tube suspension was recorded at each speed over a duration of 120 s. Equation (1) was applied to determine the flow rate of the circulating system, with the suspension process being conducted in triplicate. Subsequently, the volume of the tube suspension collected at each speed over a 120-s interval should be documented.

(1) Q=Vn−Vo120s.

In this study, Q denotes the flow velocity, Vn signifies the volume of suspension collected at each rate over a duration of 120 s, and V0 indicates the initial volume.

The shear force was determined in accordance with Eq. (2):

(2) τ=6μQa2b.

τ is the shear force, μ is the medium viscosity (0.0076 P), a is the channel height of the flow cell (0.013 cm), b is the channel width (1.0 cm), and Q is the flow velocity.

Stability testing of circulation device

A volume of 0.2 mL was collected at predetermined time intervals (ranging from 0 to 60 min, with collections occurring every 5 min) to assess the fluorescence intensity of Sample III via flow cytometry. Furthermore, triplicate measurements were performed for each time point to determine the mean fluorescence intensity at each interval. Erythrocyte suspensions, both prior to and following circulation, which had been co-incubated with sensitized GFP-E. coli, were collected and analyzed utilizing an inverted fluorescence microscope.

Detection of GFP-E. coli on the surface of porcine erythrocytes cleared by PAMs

The experiment was conducted utilizing three distinct groups: Group A, Group B, and Group C. Group A functioned as the control group, in which no PAMs were utilized. Blank slides were prepared, and the cycling detection system was constructed applying cell suspension sample III as the mobile phase, in accordance with the previously described methodology. After a duration of 60 min, the porcine erythrocyte suspension was collected. Group B served as the experimental group, in which slides containing PAMs were utilized, and the subsequent procedures were conducted in accordance with the previously outlined methodology. Group C functioned as the negative control group, utilizing slides containing PAMs in conjunction with a mobile phase composed of a suspension of porcine erythrocytes devoid of sensitized GFP-E. coli. The methodology applied was consistent with that previously outlined. The rate of reduction in surface fluorescence intensity of porcine erythrocytes, both prior to and following cycling, was calculated for groups A, B, and C using Eq. (3).

This study considers the following variables: the initial fluorescence intensity (F0) of porcine erythrocytes, the fluorescence intensity after 60 min of cycling (Ft), and the rate of reduction in fluorescence intensity (FΔ).

(3) FΔ=F0−Ft60min.

The calculation of the rate of change in fluorescence intensity before and after PAMs cycling were performed using Eq. (4) for groups B and C. In this equation, F′0 represents the initial fluorescence intensity of the PAMs at 0 min, F′t represents the fluorescence intensity after 60 min of cycling, and F′Δ represents the rate of change of fluorescence intensity.

(4) FΔ′=Ft′−F0′60min.

Effect of immunoblocking of CR1-like receptors on the surface of PAMs on GFP-E. coli transfer

The experimental groups were categorized into two distinct groups, designated as A and B. In Group A utilized slides containing PAMs, which were incubated with 5% bovine serum albumin (BSA) for 30 min, followed by incubation with a 1:50 dilution of porcine CR1-like monoclonal antibody at 37 °C for 1 h. In Group B, slides containing PAMs were incubated with 5% BSA for 30 min, and PBS was utilized in place of CR1-like monoclonal antibody (McAb). The subsequent procedures mirrored those applied in Group A. Slides containing PAMs were collected, and the fluorescence intensity of individual cells was assessed using flow cytometry.

Effect of immunoblocking of FcR on the surface of PAMs on GFP-E. coli transfer

The experimental groups were categorized into two distinct groups, designated as C and D. In Group C, slides containing PAMs were incubated with a 1:100 dilution of porcine Fc receptor monoclonal antibody (McAb) at 37 °C for 1 h. In Group D, slides containing PAMs were incubated with PBS at 37 °C for the same duration. Subsequently, erythrocyte suspensions and PAMs were collected separately following the aforementioned cycling protocol, and the fluorescence intensity of individual cells was quantified using flow cytometry.

Effect of immunoblocking of CR1-like receptors and FcR on the surface of PAMs on GFP-E. coli transfer

The experimental groups were categorized into two distinct groups, designated as E and F. In Group E, slides containing PAMs slides were treated with porcine CR1-like McAb (1:50) and FcR McAb (1:100) at 37 °C for 1 h. In Group F, slides containing PAMs were subjected to treatment with PBS under identical temperature and time conditions. Subsequent to these treatments, erythrocyte suspensions and PAMs were separately collected in accordance with the previously established cycling system. The fluorescence intensity of individual cells was subsequently quantified utilizing flow cytometry.

Observations of PAMs clearing porcine erythrocytes immune adhesion GFP-E. coli

The circulating cell suspension was co-incubated and subsequently fixed using a solution containing 0.5% and 3% glutaraldehyde, along with 1% osmium tetroxide. Samples were gradually dehydrated by acetone, followed by permeation, embedding, and sectioning techniques (80 nm). Image acquisition and analysis conducted via transmission electron microscopy.

Statistical analysis

The experimental data were analyzed using one-way Analysis of variance (ANOVA) with GraphPad Prism five software (GraphPad Software, San Diego, CA, USA), and the results were reported as the mean ± standard deviation (SD). The fluorescence intensity at each time point was analyzed, with p < 0.05 indicated a significant difference. p < 0.01 was considered extremely significant, while p > 0.05 indicated not statistically significant.

Results

Identification of porcine erythrocytes adhesion-sensitized GFP-E. coli samples from the circulatory system

Following the incubation of sensitized GFP-E. coli with porcine erythrocytes, green fluorescence was observed on the surface of the erythrocytes. The results indicated that porcine erythrocyte immunoadhesion-sensitized GFP-E. coli was successfully prepared and deemed appropriate for subsequent experimental investigations (Fig. 1). Flow cytometry analysis revealed a significant difference in the peak fluorescence intensity of porcine erythrocytes between the PBS-incubated group and the sensitized GFP-E. coli group. The sensitized adhesion group exhibited a notable rightward shift in peak fluorescence intensity, signifying an enhancement in the fluorescence intensity of the porcine erythrocytes. The results indicated that porcine erythrocytes were able to immunoadhesion to sensitized GFP-E. coli, with a mean fluorescence intensity of 14.5 ± 0.03 (Fig. 2).

Figure 1 Sample preparation of porcine erythrocytes adhering to sensitized GFP-E. coli.

(A) Shows the results of incubation porcine erythrocytes with PBS; (B) shows the results of incubation of porcine erythrocytes with sensitized E. coli with green fluorescence on the surface of porcine erythrocytes (the porcine erythrocytes and GFP-E. coli were indicated with white arrows, 100 × oil microscope, bar = 10 μm).

Figure 2 Identification of porcine erythrocytes adhesion-sensitized GFP-E. coli samples from the circulatory system.

(A) and (B) show the flow cytometry results of the control group, test group and merge results respectively. Flow cytometry analysis revealed distinct peak fluorescence intensities for porcine erythrocytes between the PBS group and the sensitized GFP-E. coli group (Figs. A, B).

Determination of mobile phase flow rate and shear

The calculations of mobile phase velocity and shear are presented in Table 1. The results indicated that the flow rate of the mobile phase within the circulatory system varies with different rotational speeds of the peristaltic pump. Research indicates (Yang et al., 2018) that the shear force of venous blood flow in its natural state is 1–6 dyne/cm2. Under experimental conditions, shear forces below 4 dyne/cm2 are classified as low shear and may result in an unstable mobile phase flow state. Consequently, in this experiment, the peristaltic pump was calibrated to operate at a speed of 4 dyne/cm2, resulting in a shear force of 5.3 dyne/cm2 for subsequent experimental procedures.

Table 1 The calculation results of flow rate and shear force.

Rotational speed (r/min)	Initial volume (mL)	Final volume (mL)	Time (s)	Flow rate (mL/s)	τ (dynes/cm2)	
1	0	0.57 ± 0.047	120	0.0047 ± 0.0004	1.30	
2	0	1.06 ± 0.047	120	0.00875 ± 0.0003	2.50	
3	0	1.70 ± 0.081	120	0.0146 ± 0.0006	4.10	
4	0	2.23 ± 0.047	120	0.0186 ± 0.0004	5.30	
5	0	2.70 ± 0.081	120	0.0225 ± 0.0007	6.40	
6	0	3.23 ± 0.120	120	0.027 ± 0.001	7.60	
7	0	3.63 ± 0.047	120	0.0303 ± 0.0004	8.60	
8	0	4.17 ± 0.047	120	0.0347 ± 0.0004	9.80	
9	0	4.73 ± 0.047	120	0.0394 ± 0.0004	11.15	
10	0	5.10 ± 0.081	120	0.0425 ± 0.0007	12.00	
11	0	5.73 ± 0.047	120	0.0478 ± 0.0004	13.50	
12	0	6.13 ± 0.047	120	0.0511 ± 0.0004	14.50	
13	0	6.70 ± 0.081	120	0.0558 ± 0.0007	15.80	
14	0	7.23 ± 0.081	120	0.0603 ± 0.004	17.10	
15	0	7.67 ± 0.047	120	0.0638 ± 0.0004	18.10	

Determination of the maximum retention time of the mobile phase of a circulating system

The mean fluorescence intensity of porcine erythrocytes was assessed at intervals of 0, 5, 10, 15, 20, 25, 30, 35, 40, 45, 50, 55, and 60 min throughout the cycle using flow cytometry (Fig. 3). A one-way ANOVA (Fig. 4) indicated that the differences in fluorescence intensity across various time points were not statistically significant (p > 0.05). This suggested that the fluorescence intensity of porcine erythrocyte suspensions remained stable during the cyclic flow from 0 to 60 min within the circulatory system, thereby demonstrating no loss of GFP-E. coli. Furthermore, the fluorescence intensity of GFP-E. coli adhered to porcine erythrocytes did not exhibit change throughout the 60-min cycling period. The porcine erythrocytes, both pre- and post-circulation, were examined utilizing an inverted fluorescence microscope. Fluorescence microscopy analysis revealed that the morphology of the porcine erythrocytes remained largely unchanged (Fig. 5).

Figure 3 Representative results from flow cytometry at various time.

The mean fluorescence intensity of porcine erythrocytes was measured at 0, 5, 10, 15, 20, 25, 30, 35, 40, 45, 50, 55, and 60 min throughout the cycle. Throughout the 60 min duration, no significant changes in the fluorescence intensity of porcine erythrocytes were observed.

Figure 4 The mean fluorescence intensity of porcine erythrocytes.

The mean fluorescence intensity of porcine erythrocytes was analyzed at 0, 5, 10, 15, 20, 25, 30, 35, 40, 45, 50, 55 and 60 min of after the post-cycle using a one-way ANOVA. The analysis revealed no statistically significant differences in fluorescence intensity across the different time points (p > 0.05).

Figure 5 Microscopic examination results of porcine erythrocyte surface.

(A) Before the cycle, (B) after the cycle, 100× oil mirror, bar = 20 μm. The revealed that the overall morphology of the porcine erythrocytes remained plump and round, exhibiting no significant alterations.

Detection of changes in surface fluorescence intensity of porcine erythrocytes in the circulatory system

Flow cytometry analysis indicated that the mean fluorescence intensity on the surface of porcine erythrocytes in Group A (control group without PAMs) was 16.83 ± 0.075 prior to cycling and 16.7 ± 0.02 following cycling. The t-test analysis indicated that the variation in fluorescence intensity of porcine erythrocytes before and after cycling was not statistically significant (p > 0.05) (Fig. 6A). In contrast, the mean fluorescence intensity on the surface of porcine erythrocytes in group B (experimental group) was 13.63 ± 0.21 before cycling and 9.06 ± 1.91 after cycling. The t-test analysis revealed that this difference was highly significant (p < 0.01) (Fig. 6B). The fluorescence intensity on the surface of porcine erythrocytes, both prior to and following cycling in group C (negative control group), was measured at 0%, with no detectable fluorescence observed. Based on the analysis using Eq. (3), the FΔ value for porcine erythrocytes was determined to be 0.002 ± 0.07 in group A, 0.08 ± 0.02 in group B, and 0 in group C. A one-way analysis of variance (ANOVA) revealed that the FΔ value in group B was significantly greater than those in groups A and C (p < 0.01) (Fig. 7). The aforementioned results indicated a reduction in fluorescence intensity on the surface of porcine erythrocytes, suggesting a corresponding decrease in the number of sensitized GFP-E. coli present on the erythrocyte surface.

Figure 6 The fluorescence intensity on the surface of porcine erythrocytes was assessed before and after circulation.

**P < 0.01.

Figure 7 Comparison of the reduction rate of erythrocyte fluorescence intensity before and after circulation.

***P < 0.001.

Detection of changes in fluorescence intensity on the surface of PAMs in the circulatory system

In Group A, no PAMs were present, and consequently, the rate of change in fluorescence intensity was not calculated. In contrast, Group B exhibited fluorescence intensities of PAMs on its surface of 0 and 3.9 ± 0.73 before and after cycling, respectively, indicating a significant increase in fluorescence intensity (Figs. 8A–8C). Meanwhile, Group C displayed fluorescence intensities of PAMs on its surface of 0 before and after cycling (Figs. 8D–8F), with no detectable fluorescence observed. Based on the analysis using Eq. (4), the F′Δ value for PAMs was determined to be 0.07 ± 0.31 in group B and 0 in group C. This indicates that PAMs have the capability to remove sensitized GFP-E. coli to which porcine erythrocytes adhere.

Figure 8 The variations in PAMs fluorescence intensity pre-circulation and post-circulation.

In Group A, the rate of change in fluorescence intensity was not calculated (no PAMs); in Group B, the fluorescence intensities of PAM on its surface before and after cycling were 0 and 3.9 ± 0.73 (A–C); in Group C, the fluorescence intensities of PAMs on its surface before and after cycling were 0 and 0 (D–F), and no fluorescence was detected.

Effect of immunoblocking of CR1-like receptor on the surface of PAMs on GFP-E. coli transfer

In Group A, PAMs were initially treated with a CR1-like monoclonal antibody, followed by interaction with sensitized GFP-E. coli adhering immunologically to porcine erythrocytes. The study observed that the mean fluorescence intensity of porcine erythrocytes was 16.00 ± 0.09 at the initial time point (0 min). In contrast, the mean fluorescence intensity of PAMs was initially 0. After 60 min of circulation, the mean fluorescence intensity of porcine erythrocytes decreased to 11.33 ± 0.09, accompanied by a leftward shift in the peak position of fluorescence intensity (Figs. 9A–9C). Conversely, the fluorescence intensity of PAMs increased to 3.57 ± 0.12, with a rightward shift in the peak position of fluorescence intensity (Figs. 10A–10C). These findings suggested a decrease in the fluorescence intensity of porcine erythrocytes, while concurrently indicating an increase in fluorescence intensity on the surface of the PAMs. In Group B, PAMs interacted with sensitized GFP-E. coli adhering immunologically to porcine erythrocytes. The initial average fluorescence intensity of porcine erythrocytes was measured 15.13 ± 0.12 at 0 time, whereas PAMs exhibited an average fluorescence intensity of 0. Following a 60 min circulation period, the average fluorescence intensity of porcine erythrocytes decreased to 8.10 ± 0.82, accompanied by a leftward shift in the peak position of fluorescence intensity (Figs. 9D–9F). Conversely, the mean fluorescence intensity of PAMs increased to 6.27 ± 0.09, with the peak position of fluorescence intensity shifting to the right (Figs. 10D–10F). This suggested a reduction in fluorescence intensity on the surface of porcine erythrocytes, while indicating an increase in fluorescence intensity on the surface of PAMs.

Figure 9 The detection of GFP-E. coli on porcine erythrocyte surface before and after cycling.

Representative flow cytometry diagrams for the CR1-like receptors blockade group (A and B), (C) the superimposed results of A and B; Representative flow cytometry diagrams for the non-blockade group (D and E), and (F) the superimposed results of D and E. The leftward shift in the peak position of fluorescence intensity.

Figure 10 The detection of GFP-E. coli on PAMs surface before and after cycling.

Representative flow cytometry diagrams for the CR1-like receptors blockade group (A and B), (C) the superimposed results of A and B; Representative flow cytometry diagrams for the non-blockade group (D and E), and (F) the superimposed results of D and E. The rightward shift in the peak position of fluorescence intensity.

Based on the analysis of Eqs. (3) and (4), the FΔ values for porcine erythrocytes and PAMs in the CR1-like receptors blockade group were 0.077 ± 0.003 and 0.06 ± 0.002, respectively. In contrast, the FΔ values for porcine erythrocytes and PAMs in the non-blockade group were 0.12 ± 0.003 and 0.10 ± 0.002, respectively. Statistical evaluation using a t-test revealed that the FΔ in the CR1-like receptors blockade group was significantly lower than in the non-blockade group (p < 0.01) (Figs. 11). The results indicated that the rate of change in fluorescence intensity for both the porcine erythrocyte surface and PAMs was significantly diminished following treatment with the CR1-like monoclonal antibody.

Figure 11 Comparison of FΔ of porcine erythrocytes and comparison of F’Δ of PAMs.

***P < 0.001.

Effect of immunoblocking of FcR on the surface of PAM on GFP-E. coli transfer

In Group C, PAMs were initially treated with a Fc monoclonal antibody, followed by interaction with sensitized GFP-E. coli adhering immunologically to porcine erythrocytes. The study determined that the initial average fluorescence intensity of porcine erythrocytes was 11.77 ± 0.50 at 0 min, whereas the average fluorescence intensity of PAMs was initially 0. After 60 min of circulation, the average fluorescence intensity of porcine erythrocytes decreased to 7.63 ± 0.21, with a leftward shift in the peak position of the fluorescence intensity (Figs. 12A–12C). Conversely, the fluorescence intensity of PAMs increased to 3.50 ± 0.80, accompanied by a rightward shift in the peak position of the fluorescence intensity (Figs. 13A–13C). The results indicated a decrease in fluorescence intensity of porcine erythrocytes, suggesting a concomitant increase in fluorescence intensity on the surface of the PAMs. In Group D, PAMs interacted with sensitized GFP-E.coli adhering immunologically to porcine erythrocytes. The study observed that the mean fluorescence intensity of porcine erythrocytes was 12.43 ± 0.17 at the initial time point (0 min), whereas the mean fluorescence intensity of PAMs was recorded as 0. Following 60 min of circulation, the mean fluorescence intensity of porcine erythrocytes decreased to 5.47 ± 0.05, accompanied by a leftward shift in the peak position of fluorescence intensity (Figs. 12D–12F). Conversely, the mean fluorescence intensity of PAMs increased to 4.90 ± 0.78, with the peak position of fluorescence intensity shifting to the right (Figs. 13D–13F). The results indicated a decrease in fluorescence intensity of porcine erythrocytes, suggesting a concomitant increase in fluorescence intensity on the surface of the PAMs.

Figure 12 The detection of GFP-E. coli on porcine erythrocyte surface before and after cycling.

Representative flow cytometry diagrams for the FcR blockade group (A and B), (C) the superimposed results of A and B Representative flow cytometry diagrams for the non-blockade group (D and E), and (F) the superimposed results of D and E. The leftward shift in the peak position of fluorescence intensity.

Figure 13 The detection of GFP-E. coli on PAMs surface before and after cycling.

Representative flow cytometry diagrams for the FcR blockade group (A and B), (C) the superimposed results of A and B; Representative flow cytometry diagrams for the non-blockade group (D and E), and (F) the superimposed results of D and E. The rightward shift in the peak position of fluorescence intensity.

Based on the analysis of Eqs. (3) and (4), the FΔ values for porcine erythrocytes and PAMs in the FcR blockade group were 0.069 ± 0.003 and 0.059 ± 0.002, respectively. In contrast, the FΔ values for porcine erythrocytes and PAMs in the non-blockade group were 0.116 ± 0.003 and 0.085 ± 0.008, respectively. Statistical evaluation using a t-test revealed that the FΔ in the FcR blockade group was significantly lower than in the non-blockade group (p < 0.01) (Fig. 14). The results indicated that the rate of change in fluorescence intensity for both the porcine erythrocyte surface and PAMs was significantly diminished following treatment with the Fc monoclonal antibody.

Figure 14 Comparison of FΔ of porcine erythrocytes and Comparison of F’Δ of PAMs.

**P < 0.01.

Effect of synergistic blockade of CR1-like receptors and FcR on the transfer of GFP-E. coli by porcine erythrocytes

In Group E, PAMs were initially treated with CR1-like monoclonal antibody and Fc monoclonal antibody, followed by interaction with sensitized GFP-E.coli adhering immunologically to porcine erythrocytes. The study determined that the mean fluorescence intensity of porcine erythrocytes was 22.03 ± 0.05 at the initial time point (0 min). In contrast, the mean fluorescence intensity of PAMs was initially 0. After 60 min of circulation, the mean fluorescence intensity of porcine erythrocytes decreased to 16.67 ± 0.25, accompanied by a leftward shift in the peak position of the fluorescence intensity (Figs. 15A–15C). Conversely, the fluorescence intensity of PAMs increased to 3.03 ± 0.05, accompanied by a rightward shift in the peak position (Figs. 16A–16C). The results indicated a decrease in fluorescence intensity of porcine erythrocytes, suggesting a concomitant increase in fluorescence intensity on the surface of the PAMs. In Group F, PAMs interacted with sensitized GFP-E. coli adhering immunologically to porcine erythrocytes. The study observed that the mean fluorescence intensity of porcine erythrocytes was 22.03 ± 0.047 at the initial time point (0 min), whereas the average fluorescence intensity for PAMs was initially 0. Following 60 min of circulation, the average fluorescence intensity of porcine erythrocytes decreased to 13.3 ± 0.28, accompanied by a leftward shift in the peak position of the fluorescence intensity (Figs. 15D–15F). Conversely, the mean fluorescence intensity of PAMs increased to 5.3 ± 0.08, accompanied by a rightward shift in the peak position (Figs. 16D–16F). The results indicated a decrease in fluorescence intensity of porcine erythrocytes, suggesting a concomitant increase in fluorescence intensity on the surface of the PAMs.

Figure 15 The detection of GFP-E. coli on porcine erythrocyte surface before and after cycling.

Representative flow cytometry diagrams for the co-blockade group (A and B), (C) the superimposed results of A and B; Representative flow cytometry diagrams for the non-blockade group (D and E), and (F) the superimposed results of D and E. The leftward shift in the peak position of fluorescence intensity.

Figure 16 The detection of GFP-E. coli on PAMs surface before and after cycling.

Representative flow cytometry diagrams for the co-blockade group (A and B), (C) the superimposed results of A and B; Representative flow cytometry diagrams for the non-blockade group (D and E), and (F) the superimposed results of D and E. The rightward shift in the peak position of fluorescence intensity.

Based on the analysis of Eqs. (3) and (4), the FΔ values for porcine erythrocytes and PAMs in the co-blockade group were 0.089 ± 0.003 and 0.05 ± 0.007, respectively. In contrast, the FΔ values for porcine erythrocytes and PAMs in the non-blockade group were 0.145 ± 0.003 and 0.088 ± 0.001, respectively. Statistical evaluation using a t-test revealed that the FΔ in the co-blockade group was significantly lower than in the non-blockade group (p < 0.01) (Fig. 17). The results indicated that the rate of change in fluorescence intensity for both the porcine erythrocyte surface and PAMs was significantly diminished following treatment with the CR1-like monoclonal antibody and Fc monoclonal antibody.

Figure 17 Comparison of FΔ of porcine erythrocytes and Comparison of F’Δ of PAMs.

**P < 0.01.

Observations of PAMs clearing porcine erythrocytes immune adhesion GFP-E. coli

The examination of the post-cycle fixed cell suspension via transmission electron microscopy revealed that the overall morphology of PAMs cells was plump and round. Observations included erythrocyte adhesion-sensitized GFP-E. coli, erythrocyte adhesion GFP-E. coli captured by PAMs, and erythrocytes detaching after sensitized GFP-E. coli was captured by PAMs, all of which were visible at the periphery of the cell membrane. These findings indicate that PAMs are capable of removing the immunoadhesion GFP-E. coli from porcine erythrocytes (Fig. 18).

Figure 18 Electron microscopic observation of sensitized E. coli on porcine erythrocytes surface captured by PAMs.

The voltage is 80 V, the scale = 2 μm, and the magnification is 8,000×.

Discussion

Since the introduction of the “erythrocyte immune system” concept in immunology (Siegel, Liu & Gleicher, 1981), significant attention has been directed towards understanding the immune functions of erythrocytes, and the research on its function has developed rapidly (Ren, Yan & Yang, 2023). Research has demonstrated that human erythrocytes are crucial innate immune cells within the blood circulation (Papadopoulos et al., 2021), widely participating in both specific and non-specific immune responses (Dunkelberger & Song, 2010). Among these, erythrocyte complement receptor 1 (ECR1) serves as the most important material basis for the immune adhesion function of erythrocytes (Birmingham, 1995), erythrocytes have the capacity to adhere to E. coli that has been opsonized by serum (Fine et al., 1980). Research has demonstrated that erythrocytes, through the mediation of CR1, are involved in the presentation of opsonized antigens, the clearance of immune complexes (ICs), and the facilitation of phagocytosis (Török et al., 2015). Research has demonstrated that immune complexes (ICs) are predominantly eliminated from circulation via the mononuclear phagocytic system (MPS) in both human and non-human primates (Schifferli & Taylor, 1989). The formation of immune complexe (IC) activates the classical complement pathway, and the degradation products resulting from complement activation enhance the sensitivity of the immune complexes (Lee et al., 2022; Mollnes et al., 1995). Following the passage of erythrocytes through the CR1 immunoadhesion sensitization immune complex (IC), hepatic macrophages bind the sensitized IC transported by the erythrocytes to their own surface, facilitating subsequent endocytosis and removal. The erythrocytes then re-enter the bloodstream; however, the CR1 levels on their surface diminish, potentially due to proteolytic hydrolysis occurring during this process (de Oliveira et al., 2014).

Building on this foundation, the current experiments further elucidated the biological process of opsonization of porcine erythrocyte surfaces by GFP-E. coli through the binding of PAMs.

We built a detection system with a closed-circulation flow chamber and a constant flow pump. By analysis of the shear force magnitude across varying rotational speeds and an assessment of potential leakage of the mobile phase during cycling (Dabagh et al., 2014; Zeng et al., 2018). Consequently, in this experiment, the peristaltic pump was calibrated to operate at a speed of 4 dyne/cm², resulting in a shear force of 5.3 dyne/cm² for subsequent experimental procedures.

The fluorescence intensity on the surface of the porcine erythrocytes and PAMs reached its plateau at approximately 55 min. Consequently, the fluorescence intensity on the surface of porcine erythrocytes was dynamically monitored over a 60 min period during the cycling of the mobile phase, with measurements recorded at 5 min intervals. The fluorescence intensity measured at 13 distinct time points underwent statistical analysis, revealing no significant differences. Indirect immunofluorescence was applied to assess porcine erythrocytes both prior to and following circulation. Fluorescence microscopy analysis revealed that the morphology of the porcine erythrocytes remained largely unchanged. Consequently, the cycling system developed in this experiment is deemed adequate for the requirements of experimental detection.

Flow cytometry analysis demonstrated a reduction in fluorescence intensity of porcine erythrocytes bound to sensitized GFP-E. coli, decreasing from 13.63 ± 0.21 to 9.06 ± 1.91, after interaction with PAMs, corresponding to a reduction rate of 8.0%. Simultaneously, the fluorescence intensity of PAMs increased from 0 to 3.9 ± 0.73, corresponding to a change rate of 6.5%. These observations suggest the transference of sensitized GFP-E. coli from the surface of porcine erythrocytes to that of PAMs. Furthermore, the decline in fluorescence intensity observed in porcine erythrocytes differed from the increase in fluorescence intensity noted in PAMs. This discrepancy may be attributed to the phagocytosis of sensitized GFP-E. coli following their binding to PAMs. Simultaneously, monoclonal antibodies targeting CR1-like receptors and Fc receptors were selected to inhibit these receptors on the surface of PAMs and to assess the effects of their simultaneous blockade. The study revealed that the activity of CR1-like receptors and Fc receptors significantly influences the immune complex (IC) involved in the clearance of porcine erythrocyte surface by PAMs. Transmission electron microscopy (TEM) was subsequently applied to observe the cell suspension following the experimental cycle. PAMs were observed to remove GFP-E. coli from the surface of erythrocyte. This finding further substantiates the interaction between erythrocytes and PAMs. Therefore, it can be inferred that when the organism is affected by exogenous factors or its molecular polymorphisms, the reduced or defective expression of molecules such as CR1-like receptors and Fc receptors on the surface of PAMs further affects the clearance of circulating immune complexes in vivo, leading to immune-complex-related diseases.

Conclusion

The CR1-like receptor and FcR on the surface of PAMs are the molecular basis for mediating the PAMs to remove the sensitized GFP-E. coli of the immune adhesion of porcine erythrocytes. This study enhanced the understanding of how porcine PAMs remove immune complexes, offering a theoretical foundation for the natural immune response of porcine erythrocytes in vivo.

Supplemental Information

Supplemental Information 1 Author Checklist.

Supplemental Information 2 Raw images for the figures.

Supplemental Information 3 Flowchart.

Additional Information and Declarations

Competing Interests

The authors declare that they have no competing interests.

Author Contributions

Yongqiang Liu conceived and designed the experiments, performed the experiments, analyzed the data, authored or reviewed drafts of the article, and approved the final draft.

Nan Wang conceived and designed the experiments, prepared figures and/or tables, and approved the final draft.

Qing Ru conceived and designed the experiments, prepared figures and/or tables, and approved the final draft.

Kuohai Fan conceived and designed the experiments, authored or reviewed drafts of the article, and approved the final draft.

Na Sun conceived and designed the experiments, authored or reviewed drafts of the article, and approved the final draft.

Panpan Sun conceived and designed the experiments, authored or reviewed drafts of the article, and approved the final draft.

Hongquan Li conceived and designed the experiments, authored or reviewed drafts of the article, and approved the final draft.

Wei Yin conceived and designed the experiments, analyzed the data, authored or reviewed drafts of the article, and approved the final draft.

Animal Ethics

The following information was supplied relating to ethical approvals (i.e., approving body and any reference numbers):

The experimental protocol was reviewed and approved by the Animal Ethics Committee of Shanxi Agricultural University (No. SXAU-EAW-2022P.TO.002014145).

Data Availability

The following information was supplied regarding data availability:

The data is available at figshare: ru, qing (2024). Raw Data. figshare. Dataset. https://doi.org/10.6084/m9.figshare.27073930.v1.

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
