# Peer review of "Assay of cardiopulmonary bypass system for porcine alveolar macrophages removing GFP-E. coli from erythrocyte surfaces"

_PeerJ, doi:10.7717/peerj.18934_

## Round 0.1 · original submission · Major Revisions

The reviewers acknowledge the importance of this paper. However, they point out that the paper has a lot of issues which should be resolved. I believe their comments are constructive and will help improve the paper. Please revise your manuscript according to the reviewers' comments.

Reviewer 1 ·

Basic reporting

The study established an in vitro device that simulates in vivo blood circulation, based on which it investigated and demonstrated the role of PAMs in removing GFP-E. coli. Furthermore, it was found that CR1 and FcR on the surface of PAMs are involved in the function of clearing sensitized complexes. However, the manuscript contains numerous confusing aspects in the presentation and description of the results, specifically as follows:

Experimental design

Major revisions:
1. In lines 91-92, why is there only one healthy rabbit? Does n=1 refer to a biological replicate of once? Additionally, is the usage of "rabbits" and "they" appropriate? Furthermore, which part of the experiment is the rabbit blood mentioned in the manuscript used for?
2. The quality of all images needs to be improved, and all figure legends should be rechecked for proper formatting and descriptions.
3. There are still some grammatical and descriptive issues in the manuscript that need improvement, and thorough proofreading of the manuscript is necessary.
4. In lines 403-405, the Western Blot and immunoprecipitation experiments mentioned are not indicated in the results. What is the reason for this?
5. A flowchart or conceptual diagram of the entire study needs to be created.
6. In lines 247-260 and 266-281, the grouping and treatment seem to be the same, but the conclusions are different. Is the grouping described accurately?
7. In lines 200-215, Why is the maximum retention time of the mobile phase measured for only 60 minutes? Will the fluorescence intensity remain stable if measured beyond 60 minutes?
8. In the in vitro simulation of blood flow, many variables should theoretically be considered beyond just the flow rate and shear force mentioned in the manuscript. Have temperature and pressure been adequately controlled, and is the ratio of dissolved oxygen and carbon dioxide in the simulated environment aligned with physiological conditions? Additionally, should other influencing factors be considered? If these other factors are not taken into account, some parts of the manuscript cannot be described as simulating physiological conditions and should limit and narrow the scope of the description. How was the pump speed of 4 rpm mentioned in line 196 determined? Was the effect at 5 or 6 rpm tested? It should be clarified whether the related studies referenced here are based on humans or pigs.
9. During in vitro simulations, the properties of fresh pig blood can change over time. What was the temperature during the in vitro simulation experiments, and were any changes in red blood cell morphology observed within the 60-minute duration? How can the influence of environmental factors during the in vitro simulation be ensured?
10. The quality of the images in Figure 3 should also be improved. The annotations for the X-axis and Y-axis should be made more visually appealing, and the format and size of the coordinate numbers need to be consistent. Formatting issues in the figure legend have been mentioned previously, so please ensure these are checked and corrected. Moreover, it is puzzling why the last two groups both show 60 minutes; the figure legend does not clarify what this means, which is quite confusing. What does the extra set of data represent? Please provide a detailed explanation. What does the additional half image after Figure 3 signify?

Minor revisions:
1. In lines 29-32, what is the relationship between the sentences here? The expression should be clear. In lines 29-33, the logical connection in the description should be rigorous and coherent.
2. In lines 54-55, the strain number of E. coli needs to be provided.
3. 3. Lines 63-65, "Super clean bench" should use professional terminology, such as Laminar flow hood, Clean bench, or Aseptic workstation.
4. In line 68, Is the amino acid sequence of the CR1 monoclonal antibody uploaded? Please provide the search number.
5. In lines 75-77, the full name and source of the reagents should be clearly stated.
6. In lines 106 and 110, Is the formatting of the formulas, correct? Please refer to the journal's author guidelines.
7. In line 113, the formatting between the sentences needs to be adjusted.
8. In lines 114-115, the units can be adjusted, clarity and conciseness are essential.
9. The phrase 'each respective time point' in line 117 is redundant, and there are many similar issues throughout the text that need to be checked one by one for language problems.
10. In line 118, the source of the software should be indicated.
11. In lines 120-121, the 'p' representing the p-value should be lowercase. There are many similar issues throughout the manuscript that need to be checked for formatting.
12. Figure 1A should label the cell names in the same way as Figure 1B. Please carefully check the formatting of the figure legends to ensure they are correct and treat them with the same attention as the manuscript text. For example, verify if the formatting of 'GFP' in the figure legend is correct.
13. The quality of the image in Figure 2 should be improved, and the formatting of the figure legend needs to be rechecked. Please treat this matter seriously.
14. Lines 195-196 should cite relevant literature.
15. Lines 202-210 should not contain duplicate data; it can be reflected in the table instead.
16. The quality of Figure 4-6 needs to be improved.
17. In lines 299-308, the group descriptions should be consistent with the group names in this section.

Validity of the findings

NA

Additional comments

NA

·

Basic reporting

In the manuscript titled "Detection of Porcine Alveolar Macrophages Removing GFP-E. coli from the Surface of Erythrocytes under Simulated Blood Circulation In Vitro," the authors utilized GFP+ E. coli, porcine erythrocytes, and porcine alveolar macrophages (PAMs) to demonstrate the efficacy of PAMs in removing GFP-E. coli from erythrocyte surfaces. This study is novel and highlights the role of CR1 in the interaction between macrophages, erythrocytes, and E. coli complexes.

Major Concerns:
From the data presented, it appears that PAM treatment reduces fluorescence intensity in erythrocytes, with an increase in PAM fluorescence after 60 minutes of cycling. However, a reduction in GFP signal in erythrocytes does not directly demonstrate that PAMs remove GFP-E. coli from erythrocyte surfaces. It is essential to clarify whether PAMs are phagocytosing surface-adhered E. coli or whether there is efferocytosis of E. coli-adhered erythrocytes. Additionally, in lines 280, 314, and 350, the authors mention, "the fluorescence intensity on the surface of PAMs was significantly reduced," suggesting that E. coli was attached to the PAM surface. This interpretation is technically incorrect; PAMs should have phagocytosed the E. coli, and there are no microscopic images showing E. coli adhered to the PAM surface. These issues must be addressed either by providing appropriate evidence or by revising the text.

Furthermore, the study does not examine differences in the absolute number of erythrocytes before and after cycling, with and without PAMs. Including this data would clarify if PAMs are indeed phagocytosing erythrocytes.

Minor Comments:
The rationale for selecting alveolar macrophages for studying these interactions is unclear. Please discuss the reasoning behind using a tissue-resident macrophage instead of macrophages such as bone marrow-derived macrophages.

In lines 88–89, provide the methodology and references for the separation and purification of PAMs and porcine erythrocytes.

In lines 154–158, information regarding Group B is duplicated. Please remove the redundant sentence.

Results are presented in 20 figures. I strongly recommend combining all figures under each title into a single figure to facilitate easier comprehension for readers.

Consider excluding graphs that do not present notable values. For instance, Figure 5C could be summarized in text rather than in a separate figure.

Data redundancy: Lines 204–210 repeat information already represented in text, Figure 3, and Table 3. This should be streamlined to avoid redundancy, retaining only one form of representation. This suggestion also applies to the text describing Figure 5 and Table 4.

Experimental design

No comment

Validity of the findings

No comment

Reviewer 3 ·

Basic reporting

In the present manuscript entitled “Detection of porcine alveolar macrophages removing GFP-E-coli from the surface of erythrocytes under simulated blood circulation in vitro”, the author proposed the role of porcine erythrocytes in immune complex clearance by examining their interactions with alveolar macrophages and fluorescent-labeled E. coli under simulated in vitro blood simulation. Although this is an interesting study, there are several questions that need to be addressed before it is suitable for publication. Therefore, a major revision is necessary to improve the quality of the current manuscript.

Language
• It is strongly recommended that the authors consider using professional language services to improve the clarity and quality of the manuscript
Abstract
• The authors should elaborate on the in vitro blood simulation system and optimized parameters. It would be beneficial to highlight the major findings of the study to offer readers a glimpse of the results.
Introduction
• It was mentioned that erythrocytes transport ICs to the monocyte phagocytosis system via CR1, it would be beneficial to describe how CR1 facilitates this process. This could include a summary of current theories on the molecular mechanisms of IC capture, as well as known CR1 interactions with macrophages to underline the gaps in knowledge that remain to be addressed.

Experimental design

Methods
• Methods in general require major language improvement. For example in Line 84-85 “carotid artery is broken” should be “carotid artery was broken”
• Lines 93-100 need revision to ensure that past tense is used to describe the experiments that were conducted.
• The names of bacteria (e.g. E. coli) should be italicized, but they are not consistently formatted throughout the manuscript. Please revise for consistency
• Line 113. “Stability testing of circulation device” is a sub-title and not part of the method description. Please revise.
• Line 148-149 may not be essential to the manuscript. The authors have mentioned the same information in Line 177-179.

Validity of the findings

Results
• In Table 2, the authors should use a standardized numerical decimal format or preferably scientific notation, especially in columns – “Final volume” and “Flow rate”
• In Table 3, it is unclear what do the middle columns mean. Replicates? Kindly revise with appropriate labelling.
• Lines 204-210 may not need to list all the values. Instead, highlight the most significant values and explain their contribution to the overall research findings.
• There are too many figures in this manuscript, which may overwhelm readers. Please consider reducing the figure numbers by combining related figures where possible. For example, Figures 1 and 2 could be presented together, as well as Figures 9 and 10, and Figures 13 and 14, etc.
• The authors poorly discussed about Figure 6 which indicates that the figure may not be significant to the overall findings. Please consider removing the figure or moving it to the supplementary section.
• Line 237-241 are repeating the caption for Figure 8. Please provide an appropriate description of results.
• Line 335. “group A” should be “Group A”
Discussion
• Please ensure that the value is stated consistently throughout the manuscript. On line 376, it is listed as 4 rpm/min, but it has also been mentioned as 4 r/min. Kindly revise.
• Line 379-381. Please provide a more robust justification for validating the system, potentially including references to previous work to support your rationale.
• Line 405 mentions that different assays were used to validate the findings, but the results from these assays are not reported in the manuscript. The figures primarily focus on flow cytometry.
• Line 411. Please cite the previous studies used to validate current findings.
• Line 427. Please provide citations for the previous studies.
Conclusion
• Conclusion lacks a clear overview of the current study and its impact on the state of the art. Please revise by providing a more impactful summary of the study's findings and potential directions for future work.

---

## Round 0.2 · Minor Revisions

The reviewers have acknowledged your revisions. However, one of the reviewers still left some comments. I also think these comments can help you improve your manuscript. Please revise your manuscript according to the comments.

·

Basic reporting

No comment

Experimental design

Well defined experimental design

Validity of the findings

No comments

Reviewer 3 ·

Basic reporting

In the current manuscript titled “Assay of Cardiopulmonary Bypass System for Porcine Alveolar Macrophages Removing GFP-E. coli from Erythrocyte Surfaces,” the authors have made substantial improvements following the review. The manuscript is suitable for acceptance after addressing the following minor revisions:

Experimental design

Lines 84–88: The language in this section should be made consistent with the rest of the text.

Validity of the findings

The quality of Figure 3 needs to be enhanced because most parts appear blurry.
Figure 5 looks like two figures overlapping.
Figure 18 mentions a red box, but only a grayscale image is provided.
It is also highly recommended to reduce the overall number of figures, as suggested in the initial review.

---

## Round 0.3 · accepted · Accept

The reviewers are now satisfied with the revised manuscript. This paper will be published.

Reviewer 3 ·

Basic reporting

No comment

Experimental design

No comment

Validity of the findings

The authors have addressed all concerns